# Evaluation of Total Phenolic and Flavonoid Contents, Antibacterial and Antibiofilm Activities of Hungarian Propolis Ethanolic Extract against *Staphylococcus aureus*

**DOI:** 10.3390/molecules27020574

**Published:** 2022-01-17

**Authors:** Sarra Bouchelaghem, Sourav Das, Romen Singh Naorem, Lilla Czuni, Gábor Papp, Marianna Kocsis

**Affiliations:** 1Department of General and Environmental Microbiology, Institute of Biology, University of Pécs, Ifjúság Str. 6, 7624 Pécs, Hungary; bouchelaghem24sarra@gmail.com (S.B.); romen.naorem@hotmail.com (R.S.N.); czuni.lilla@gmail.com (L.C.); pappgab@gamma.ttk.pte.hu (G.P.); 2Department of Laboratory Medicine, Medical School, University of Pécs, Ifjúság Str. 13, 7624 Pécs, Hungary; pharma.souravdas@gmail.com; 3Department of Plant Biology, Institute of Biology, University of Pécs, Ifjúság str. 6, 7624 Pécs, Hungary

**Keywords:** propolis, *Staphylococcus aureus*, MRSA, antimicrobial, MIC value, interaction, biofilm, cell viability

## Abstract

Propolis is a natural bee product that is widely used in folk medicine. This study aimed to evaluate the antimicrobial and antibiofilm activities of ethanolic extract of propolis (EEP) on methicillin-resistant and sensitive *Staphylococcus aureus* (MRSA and MSSA). Propolis samples were collected from six regions in Hungary. The minimum inhibitory concentrations (MIC) values and the interaction of EEP-antibiotics were evaluated by the broth microdilution and the chequerboard broth microdilution methods, respectively. The effect of EEP on biofilm formation and eradication was estimated by crystal violet assay. Resazurin/propidium iodide dyes were applied for simultaneous quantification of cellular metabolic activities and dead cells in mature biofilms. The EEP1 sample showed the highest phenolic and flavonoid contents. The EEP1 successfully prevented the growth of planktonic cells of *S. aureus* (MIC value = 50 µg/mL). Synergistic interactions were shown after the co-exposition to EEP1 and vancomycin at 10^8^ CFU/mL. The EEP1 effectively inhibited the biofilm formation and caused significant degradation of mature biofilms (50–200 µg/mL), as a consequence of the considerable decrement of metabolic activity. The EEP acts effectively as an antimicrobial and antibiofilm agent on *S. aureus*. Moreover, the simultaneous application of EEP and vancomycin could enhance their effect against MRSA infection.

## 1. Introduction

*Staphylococcus aureus* is present in the environment such as air and food and is also found in the nose, ear, throat, and the skin of healthy individuals as a commensal bacterium [1]. However, *S. aureus* can cause a wide range of infections in the blood, skin, and wounds of its host [2]. It is a major human pathogen responsible for causing various community-onset and hospital-acquired infections that result in significant morbidity and mortality [3,4]. In the past decades, rapid resistance developed against the β-lactam antibiotics via horizontal transfer of resistance determinants encoded by mobile genetic elements or by mutations in chromosomal genes [5]. Even though vancomycin is one of the most effective antibiotics that act against methicillin-resistant *Staphylococcus aureus* (MRSA) infections. However, due to improper use of antibiotics and acquisition of antibiotic-resistance genes, vancomycin-resistant MRSA (VR-MRSA) strains have emerged as well [3,4,6]. The potent biofilm-forming ability of *S. aureus* on host tissues and medical implants leads to chronic infection development [7]. Biofilm is a collection of sessile microbial communities encapsulated by an extracellular polymeric substance (EPS) matrix consisting of exopolysaccharides, proteins, lipids, minerals, and nucleic acids [8]. The EPS of biofilm communities have various roles in both the structure and function, such as promoting the adhesion of the microorganism, offering structural stability and nutrient availability [1]. Importantly, the EPS enhances biofilm tolerance to antimicrobials and immune cells and promotes pathogenesis [1,8]. The emergence of MRSA and its biofilm elimination are challenging issues in antibiotic therapy [9]. The development of alternatives to antibiotics for the inhibition and eradication of biofilm formation is one of great concern. Thus, the use of natural compounds, such as propolis would be one of the good alternative ways to address the challenge issues of antibiotic treatments [10]. Propolis is a resinous product collected by honeybees from various plant origins. Generally, raw propolis is composed of 50% plant resins, 30% beeswax, 10% aromatic oils, 5% pollen, and 5% organic and inorganic compounds (including amino acids, vitamins, and minerals) [11]. The main components of propolis are flavonoids and phenolic acids, which dissolve easily in organic solvents usually in ethanol [12]. The biological properties of propolis extract are mostly related to polyphenols. Propolis has been used as a folk medicine due to its diverse biological properties such as antibacterial, antifungal, antiviral [13,14], antioxidant, anti-inflammatory, antitumour, antimutagenic [15], antiradiation, wound healing [16], and food preservative [17,18]. Recently, researchers have focused more on the therapeutic properties of propolis from around the world, although this remains an unclear topic [13,19,20]. To acquire further understanding of this field, this study aimed to determine and compare the total phenolic and flavonoid contents of ethanolic extracts of Hungarian propolis (EEP). Furthermore, in vitro antibacterial activity was investigated alone and in combinations with oxacillin, cefoxitin, and vancomycin on methicillin-resistant and methicillin-sensitive *S. aureus* strains. On the other hand, the antibiofilm activity of EEP against planktonic cells and mature biofilms of *S. aureus* strains were examined.

## 2. Results and Discussion

### 2.1. Determination of the Total Phenolic and Flavonoid Contents

Propolis is a resinous product collected from various plant sources such as birch, poplars, and conifers. Due to the varied origin and plant sources of propolis, the contents of phenols and flavonoids are mostly variable [21,22]. In Hungary, the main sources of propolis originate from the bud secretions of poplar (*Populus* spp.) and birch (*Betula* spp.), which are sources of resin [13]. Propolis is a sticky substance having different colours from dark-brown to yellow due to the change of the chemical composition, especially flavonoid and phenolic compounds, that is related to the changing of vegetation and plant source (vary depending on the area from where propolis is collected whether different or even from the same region) [21]. In the current study, the total polyphenolics content (TPC) and total flavonoids content (TFC) of the EEP samples had been measured to determine the concentrations of the main components of the propolis samples (Table 1).

The result of TPC values was found in the range of 10.4–71.1 mg of gallic acid equivalent (GAE)/g, and TFC values were found in the range of 33.8–273.2 mg of catechin equivalent (CAE)/g. The EEP1 sample represented the highest concentration of phenols and flavonoids and showed 6.8 to 8.1 times higher contents than that of the EEP6 sample. Among the studied samples, EEP6 sample showed the lowest values of TPC and TFC. Since all the samples were collected in the same season and extracted by the same method, the diversity of the vegetation between the regions is likely the reason for the significant difference in the polyphenolic contents. Our propolis sample showed a phenolic content about 4 times lower than other samples collected from different regions of Hungary, which were in the range of 104.6 and 286.9 mg GAE/g [23]. The concentration of phenolic or flavonoid contents was proportional to the colour intensity. In agreement with Machado and co-workers [24], the visual comparison revealed that the light yellow colour represents low concentration, while the dark colour represents high concentration of TPC and TFC (data not shown). The Mexican samples showed a high content of phenols from 68 to 500 mg caffeic acid equivalents (CAE)/g, and flavonoids from 13 to 379 mg quercetin equivalents (QE)/g. This huge difference is due to the different locations of the samples collected [25]. The TPC of different propolis samples (Bolivia, Poland, and Romania) ranged from 43 to 343 mg/g, and TFC ranged from 5 to 144 mg/g [17,26,27]. The TFC of propolis was found to vary from 135.93 to 326.10 µM QE/g. Meanwhile, the TPC varied from 28.57 to 55.16 µM GAE/g. The contents of Malaysian propolis are influenced by the vegetation at the site of the collection as well as the species of stingless bees. Different origin of propolis contains different components thus determining its properties [28]. TPC of EEP samples varied between 27.5 and 199.7 mg GAE/g, the TFC of samples varied between 30.7 and 302.94 mg QE/g. The high concentration of TPC and TFC in Turkish propolis may be a result of the local flora and mild climate conditions [29,30]. Folin-Ciocalteu and AlCl_3_ methods were used to determine the TPC and TFC of Venezuelan propolis samples, respectively. The results demonstrated a wide range of TPC between 19.1 and 107 mg GAE/g. In addition, TFC varied between 2.6 and 8.5 mg QE/g [31]. The ethanolic extracts of Brazilian propolis were presented a TPC of 1.26 and 3.87 mg GAE/g, the TFC was 0.14 and 0.15 mg QE/g. These propolis samples showed significantly less concentration of TPC and TFC, therefore less antimicrobial activity. TPC was different between the EEP samples depending on the bee species, as well as other factors [32]. The TPC and TFC of various solvents extracts of Algerian propolis ranged from 0.81 to 8.97 mg GAE/g and from 0.57 to 3.53 mg QE/g, respectively [33]. These samples were lacking in phenols and flavonoids compared to the other studies. However, Mohtar et al. [31] was suggested that the lack of significant content of phenols and flavonoids in EEP does not indicate that its antimicrobial activity is weak. The ethanolic extract of Polish propolis exhibited significantly low TFC ranged from 11.01 to 15.71 mg QE/g compared to our samples. On the other hand, the TPC was between 76.03 and 105.29 mg CAE/g [34]. Wieczy and co-authors used solvents of ethanol and hexane to prepare Polish propolis extracts, the TPC results in the examined samples ranged from 14.59 to 150.8 mg GAE/g, the highest concentration of phenols was found in ethanolic extraction [35]. The samples of Croatian propolis presented TPC values in the range of 10–220 mg GAE/g. Large variability in TFC was also observed ranging from 5 to 50 mg CAE/g [36]. Similarly, the TPC of Palestinian propolis ranged from 9.62 to 124.94 mg GAE/g, and TFC ranged from 1.06 to 75.31 mg QE/g [37]. One of the Brazilian propolis sample showed higher TPC of 482 mg GAE/g [12], while the highest TFC was 523 mg/g in Turkish propolis [38]. It has to be noted that the mode of extraction highly determines the composition of the end product, even though the biological potentialities of propolis are determined by the contents of total phenolics and flavonoids, these two parameters were agreed upon and widely applied in other studies [12,39]. Compared with previous studies, our EEP samples presented very high concentrations of flavonoids except for sample EEP6. The content of phenols was in agreement with other findings. All studies indicate that the composition of propolis strongly depends on the flora at the collection site and the bee species, as well as the solvents used for extraction and the season of collection.

### 2.2. Antibacterial Susceptibility Test

An antibiotic susceptibility test was performed against the *S. aureus* strains using oxacillin, cefoxitin, and vancomycin. The two MRSA clinical isolates (SA H23 and SA H24) were found resistant to oxacillin, and cefoxitin, but sensitive to vancomycin at 1 µg/mL (Figure 1). The reference ATCC 29213 strain was susceptible to the above antibiotics and showed MIC values at 0.25 µg/mL, 4 µg/mL, and 1 µg/mL, for oxacillin, cefoxitin, and vancomycin, respectively. The reference ATCC 700699 strain was found resistant to oxacillin, and cefoxitin; however, this strain showed intermediate resistance to vancomycin with MIC value equal to 8 µg/mL. It was reported that MRSA acquired the *mec*A gene that is present within the Staphylococcal Chromosomal Cassette *mec* (SCC*mec*) and reduced the binding affinity of β-lactam antibiotics (methicillin, oxacillin, cefoxitin, etc.) on the peptidoglycan layers of *S. aureus* [40]. Slightly different, vancomycin is a conventional glycopeptide that inhibits the late stage of cell wall biosynthesis in *S. aureus* and other Gram-positive microorganisms, by binding to the C-terminal (the D-Alanyl-D-Alanine) residue of the peptidoglycan [41]. Propolis has several possible mechanisms associated with antibacterial activity, such as inhibition of cell division, collapsing microbial cytoplasm cell membranes and cell walls, enzyme inactivation, bacteriolysis, and protein synthesis inhibition or DNA damage [10,22]. In this study, the anti-staphylococcal activity of the EEP1 sample was tested. Accordingly, all the strains were very sensitive to EEP1 with a MIC value at 50 µg/mL (Figure 1). Previous studies reported that the antibacterial activity of EEP varies depending on many factors, such as the type of propolis, the extraction method, and the method of testing on bacterial susceptibility [42,43]. The Brazilian propolis showed a very broad range of MIC values from 31.2 µg/mL to >1024 µg/mL against *S. aureus* strains [42,43]. The MIC value of the alcoholic extract of Iranian propolis was 150 µg/mL for *S. aureus*, while the MIC value for *Streptococcus mutans*, *Enterococcus faecalis*, and *Lactobacillus acidophilus* were in the range of 300–600 µg/mL. The study investigated the ability of propolis to be used in mouthwash to avoid the other popular solution, which has serious side effects such as chlorhexidine. The real-time polymerase chain reaction (RT-PCR) examination on salivary specimens of rats showed a significant reduction of oral pathogens growths after the use of mouthwash prepared from MIC value (150 µg/mL) of propolis [44]. Similarly, the growth of oral pathogens such as *S. aureus*, *Escherichia coli*, and *Candida albicans* was inhibited at MIC value ranging from 0.15% to 0.25% (*w*/*v*) of propolis ethyl alcohol extract [45]. On the other hand, the antimicrobial activity of Romanian propolis extracts were used against *S. aureus*, *E. coli*, *C. albicans*, *Bacillus cereus*, and *Pseudomonas aeruginosa*. Vică et al. reported that propolis aqueous extracts showed 8 to 11.2 mm diameter of inhibition zone at a concentration of 10 mg/mL [46]. Numerous studies confirmed the high antimicrobial potential of propolis using in vitro and in vivo assays against some important pathogens. The effect of propolis and its chemical composition is varied considerably according to the geographic area, the bee species, and the method used to obtain the extract [46,47]. Antimicrobial activity of Polish propolis against *S. aureus* and *E. coli* bacteria, and fungal species *Candida krusei*, *Mucor mucedo*, *Alternaria solani*, and *Colletotrichum gloeosporioides* was assessed using the disk diffusion method. The EEP showed inhibitory activity against all tested strains, *S. aureus* was more sensitive with 20.24–27.07 mm diameter of the inhibitory zone at a concentration of 2 mg. This study revealed that the diversity of the inhibitory activity of extracts not only depends on the used method of extraction but also on the treated microorganism species [34]. The antimicrobial activity of Croatian propolis samples was revealed against *S. aureus* bacteria and *C. albicans* yeast. Most of the samples did not display activity against *E. coli*. In addition, the MIC values were slightly elevated, ranging from 0.391 to 12.5 mg/mL on Gram-positive bacteria and yeast. The result of susceptibility test is considered to have weak therapeutic activity compared to others EEP [36]. The study of Daraghmeh and Imtara investigated the antibacterial activity of Palestinian propolis against multidrug-resistant clinical isolates, including *S. aureus*, *E. coli*, *P. aeruginosa*, and *Streptococcus faecalis*. The antibacterial effect of EEP was detected against both Gram-positive and Gram-negative pathogens with MIC values ranging from 0.01 to 5 mg/mL. Mostly, the minimum bactericidal concentrations (MBC) were equal to MIC values [37]. The ethanolic extracts of Brazilian propolis were prepared by an extraction method similar to that used in our study. The antimicrobial activity of EEP was evaluated against *S. aureus* and MRSA, using the microdilution method. The MIC value was determined in the range of 1 to 6 mg/mL, it is considered to be significantly less effective compared to our EEP1 results. However, treatment of *S. aureus* for 4 h caused significant leakage of cell constituents, which compromised the integrity of the bacterial cell membrane [32]. EEP and its fractions exhibited a wide spectrum of antibacterial and antifungal activities against Gram-positive (*S. aureus*, *S. mutans*), as well as Gram-negative (*E. coli*, *Citrobacter freundii*, and *Proteus mirabilis*), and *C. albicans* at 100 and 200 µg/mL. This study confirmed the safety of most samples when human gingival fibroblasts (HGFs) were incubated with 10 μg/mL and 100 μg/mL EEP in vitro. However, some samples at concentrations of 500 μg/mL and 1000 μg/mL induced a cytotoxic effect resulting in decreased mitochondrial activity of HGFs [35]. Potent anti-MRSA activity was demonstrated by ethanolic extracts of Turkish propolis at 952.5 μg/mL [29]. Moreover, the ethanolic extracts of Hungarian propolis were examined on different bacterial strains including *S. aureus* using an agar well diffusion assay. All studied bacteria showed bactericidal effects at 200 μg/mL with a diameter of 12 to 22.5 mm of the inhibitory zone, the antimicrobial activity of EEP was independent of the bacterial species [23]. Australian propolis effectively inhibited the growth of the MRSA strain that showed high resistance to 50 μg/mL gentamicin, it exhibited MIC value and MBC at 900 μg/mL EEP. Furthermore, DNA leaks were detected by these samples. Scanning electron microscopy confirmed that EEP damaged cell integrity including the cell wall and membrane [48]. The ethanolic and aqueous extracts of Iranian propolis showed bacteriostatic and bactericidal efficacy against oral strains at 250 and 500 μg/mL. These results suggest that EEP may be more useful in controlling caries development [49]. On the other hand, the high anti-staphylococcal activity of EEP was observed for samples collected from Taiwan, Turkey, Oman, and Ireland, with MIC values at 3.75, 8, 42, and 80 µg/mL, respectively [19,50,51,52]. Nevertheless, the ethanolic extract of Hungarian propolis can be presented as one of the most effective samples to use against the planktonic cells of clinical MRSA isolates.

### 2.3. Killing Effect of EEP1 in Combination with Antibacterial Drugs

As a result of the high rate of infection with drug-resistant bacteria over the past few decades, efforts have been intensified not only to discover new antibiotics but also to find new strategies to fight the infection [53]. The use of combination therapies between two pre-existing drugs is a promising alternative therapy, whereby the effectiveness of the treatment is enhanced at the reduced concentration of the two drugs [6,54]. The drug’s association with phenolic compounds could enhance the activity of common antibiotics against a range of resistant pathogens [53]. There has been no comparison between the use of oxacillin, cefoxitin, and vancomycin alone and in combination with EEP1 in the treatment of *S. aureus* at higher inoculum size. We used 10^8^ CFU/mL to simulate an organism density that is often associated with many infections. Staphylococcal infection often results in a high bacterial density (10^8^ to 10^10^ cells/g of tissue) [55]. The MIC value may vary according to the size of the inoculum used especially with some β-lactam antibiotics [56]. The increases in bacterial inoculum from 10^5^ to 10^8^ CFU/mL raised the MIC value of EEP1 from two to eight-fold (Figure 1 and Table 2). MRSA, SA H23 and SA H24 strains were resistant to oxacillin, and neither MIC_50_ nor MIC_80_ were observed at 10^5^ nor 10^8^ CFU/mL. MSSA shown MIC_50–80_ of oxacillin at 0.125–0.25 µg/mL on 10^5^ CFU/mL and 1–4 µg/mL on 10^8^ CFU/mL. The results of EEP1 showed that MIC_80_ equals 25 μg/mL in all the strains with 10^5^ CFU/mL, while it was increased significantly (50 μg/mL to more than 200 μg/mL) with 10^8^ CFU/mL. Further, as indicated in Table 2, the FICI values of chequerboard microdilution result of two-drug combinations between EEP1 and antibiotics (oxacillin, cefoxitin, and vancomycin) demonstrated synergistic combinations with all the antibiotics against the sensitive strain ATCC 29213, while the resistant strains were shown synergistic effect only with the vancomycin. However, The MIC values of all the antibiotics showed a significant reduction in case of interaction with propolis on all the strains, except the interaction with oxacillin on the strain SA H23 which was indifferent. Although, these concentration combinations did not show complete inhibition due to the high cell number (Table 2). The interaction properties between EEP and antibiotics on *S. aureus* have been described by previous studies, the results of Grecka and co-workers in 2019 [57] were mentioned synergistic interaction between EEP and antibiotics (amikacin, kanamycin, gentamicin, tetracycline, and fusidic acid), which are acting on the inhibition of protein synthesis. Other chemotherapies, which mostly act on inhibiting the protein synthesis (clindamycin, tetracycline, tobramycin, trimethoprim-sulfamethoxazole, and linezolid) were combined with EEP using disk diffusion assay, thus the sensitivity of *S. aureus* significantly increased. The interaction with cefoxitin also had a positive effect but was not significant compared to the absence of EEP [54]. It was also reported that the Polish propolis sample had shown additive interaction with oxacillin [57]. The present study showed that the combination of EEP with vancomycin might boost the activity to reduce the cell wall synthesis. Such similar findings on synergistic effects between EEP and antibiotics acting on cell wall biosynthesis were reported [19,58]. The synergism between EEP and five drugs (chloramphenicol, gentamicin, netilmicin, tetracycline, and vancomycin) was observed by E-test and disk diffusion methods [59]. Moreover, the study of Surek et al. has been evaluated the antibacterial effects and interaction of Brazilian propolis with antibacterial agents, using the broth microdilution method and chequerboard tests, respectively. All EEP showed potent antibacterial effects on MSSA and MRSA with MIC values in the range of 250–500 μg/mL. In addition, a time-kill test was performed to confirm the results obtained by the chequerboard test. EEP samples showed a promising synergistic effect with gentamicin against MRSA at 62.5 μg/mL EEP and 0.83 μg/mL gentamicin after 18 h incubation. None of the extracts exhibited synergism with oxacillin and vancomycin against MRSA. Thus, the results indicate that EEP is safe and effective, and can reduce the resistance to gentamicin and the occurrence of its toxic effects [47]. Most of the studies conclude a synergistic interaction between EEP and the drugs that interfere with protein synthesis on the cells. Furthermore, β-lactams and vancomycin antibiotics could positively act with propolis on the cell wall of *S. aureus* strains. It was reported that the type of the EEP and antibiotics interactions is influenced by the experimental method and depends on one strain to another, due to the presence or absence and the type of SCC*mec* carried by the cell [54,57,59,60]. Such a similar finding was observed in our present study, in which SA H23 and SA H24 strains carried SCC*mec* type IVa and type II, respectively [61]. While the reference strain ATCC 29213 has no SCC*mec*, and ATCC 700699 harboured SCC*mec* type IVa, this might result in the variation of the interaction of EEP with antibiotics (Table 2).

### 2.4. Effect of EEP1 on Biofilm Formation

Biofilm formation is an important virulence factor, characterized by the attachment of multilayered cells to abiotic and biotic surfaces [62]. *S. aureus* biofilm is developed in four stages: attachment, microcolony formation, maturation, and detachment. Naturally, the biofilm cells are more resistant to antibiotics than planktonic cells, because biofilm cells have few metabolic activities and fewer cell divisions [62]. Cells in the biofilm are embedded in an extracellular polymeric substance matrix comprising extracellular DNA, proteins, and polysaccharide intercellular adhesin (PIA) [9]. The EPS matrix supplied mechanical stability of biofilms and adhesion to surfaces. Moreover, the PIA contributes to the structural integrity of biofilms during the accumulation phase of biofilms formed by certain staphylococcal strains [63]. The PIA is the important component of the biofilm matrix which is produced and secreted by the proteins encoded in the intercellular adhesion operon (*ica*ADBC). It was demonstrated that there was a relationship between phenotypic biofilm formation and the presence of *ica*A and *ica*D genes [63]. However, it was reported that not all *ica*-positive isolates produce strong biofilm [61] and biofilm formation of *S. aureus* strains involves various *ica*-independent factors [64]. The crystal violet staining was applied to investigate the effect of EEP1 on *S. aureus* biofilm formation. The result was interpreted according to the ODc that was calculated to separate the growth of the biofilm at different concentrations of EEP1 into 4 categories: strong, moderate, weak, and no biofilm formation (Figure 2).

Biofilm formation is influenced by a variety of conditions such as environment, availability of nutrients, and above all the presence of the regulatory genes and their expression [65]. All the tested *S. aureus* strains were strong biofilm formers in the absence of EEP1. However, the biofilm formation was significantly inhibited in the presence of 100–200 µg/mL EEP1. The study by Wojtyczka and co-workers presented the inhibition of *Staphylococcus epidermidis* biofilm with EEP in the range of 0.78 to 1.56 mg/mL after 24 h incubation [66]. Another study found that EEP has the ability to impair *P. mirabilis* biofilm in the range of 25–100 mg/mL [16]. Ethanolic extract of Italian propolis has shown the ability to reduce no more than 65% of the biofilm biomass of *P. aeruginosa* at 100 µg/mL after 24 h treatment. On the other hand, the viability of sessile bacteria was diminished by 42% at the same concentration [67]. The minimum biofilm inhibitory concentration (MBIC) of EEP1 were 50 µg/mL for MRSA ATCC 700699, 100 µg/mL for the two MRSA clinical isolates (SA H23 and SA H24), and 200 µg/mL for MSSA ATCC 29213. Interestingly, MRSA biofilm was the most sensitive to propolis treatment. According to the World Health Organization (WHO), MRSA is not necessarily more dangerous than MSSA. However, MRSA has a higher mortality rate, as it is related to bacteraemia infection more than MSSA. While MSSA can also be mortal in the healthcare field especially for infants. In general, MRSA is a very common cause of hospital-acquired infections, whereas MSSA tends to be linked with community-acquired infections. No statistical difference was found between MSSA and MRSA concerning all biofilm-coding genes (*ica*A, *ica*B, *ica*C, and *ica*D) [63]. Possibly the EEP1 inhibits biofilm formation in MSSA via a mechanism that differs from that responsible for the resistant strains [62,63,65]. In our previous experiments, low concentrations of EEP1 caused an increase of the biofilm-forming ability on *Bacillus clausii* from weak to strong biofilm, regarding the simultaneously decreased swarming motility and increased autoaggregation [68]. However, in the present study, we observed a concentration-dependent monotonous inhibition of biofilm formation.

### 2.5. In Vitro Effect of EEP1 on the Eradication of Mature Biofilm

It is considered that biofilms contribute to more than 80% of all infections in humans. The formation of biofilms by MRSA and MSSA strains is an important virulence factor, affecting its persistence in both the environment and the host organism, as bacterial cells in biofilms show increased resistance against conventional antimicrobial treatments and host immune factors [65]. It is cumbersome to remove the mature biofilms and reduce the growth of dormant bacteria inside biofilm matrix, due to the difficulty of drug penetration into the biofilm. It has been well characterized that bacteria in biofilms can tolerate up to 10–1000 times higher concentrations of antibiotics than planktonic bacteria [9]. In this assay the cultures of *S. aureus* were grown for 24 h at 37 °C, then the formed biofilm was treated with various EEP1 concentrations for 16 h, and the biofilm eradication was detected by crystal violet colorimetric assay. The MBEC_50_ was defined previously as the concentration that causes a 50% reduction in the biofilm metabolic activity [69]. The EEP1 significantly enhanced the biofilm degradation in each strain and showed MBEC_50_ values of 15, 18, 48, and 52 µg/mL against MRSA ATCC 700699, SA H23, SA H24, and MSSA ATCC 29213, respectively. The biofilm of MSSA and SA H24 strains showed more resistance to the EEP1; however, the thickness of biofilms was degraded at 200 µg/mL of EEP1 by 47% and 87%, respectively. The most sensitive biofilm was observed in the case of MRSA ATCC 700699 and SA H23 strains, where the degradations of the 24 h-old biofilms were 88% and 71%, respectively, after treatment with 50 µg/mL of EEP1. In contrast, unexpected growth of the biofilm biomass of the same strains was observed in the presence of high EEP1 concentration (200 µg/mL) (Figure 3). Such a similar result was reported in the biofilm of *S. epidermidis*, it was further suggested that the efficiency of propolis can be reduced over time. In addition, after 24 h the propolis stimulates biofilm formation and added that the high concentration of EEP could be used as a nutrient by bacteria for its proliferation [66].

Wang et al. found that treatment of MRSA biofilm with 1/2 MIC, MIC (900 μg/mL), and 2 MIC values of EEP, caused a significant decrease of the cellular activity, using the XTT reduction assay. Moreover, a significant decrease in biomass of MRSA biofilms was detected after treatment with the previous concentrations, by crystal violet assay. Thus, EEP not only inhibited the planktonic cell growth but also affected the adhesion on a solid surface [48]. Furthermore, *S. aureus* biofilm was effectively eliminated from the prosthetic materials with 10% Brazilian green propolis alcohol solution after immersion for 5 min [70]. The biofilm eradication was found to be significantly influenced by the solvent used for the extraction, the bacterial strains tested, and the origin of the propolis samples. The crystal violet assay was used to determine the antibiofilm activity of Algerian propolis on *S. aureus* and MRSA bacteria at 300 μg/mL. Petroleum ether extracts showed the highest activity up to 80% reducing MRSA biofilm [71]. The researchers suggested that EEP could be used to treat chronic wound infections caused by *P. mirabilis*. As a result of the ability of EEP to inhibit and reduce the biofilm of *P. mirabilis* at 25–100 mg/mL. However, accurate determination of the appropriate concentration is very important [16]. Noteworthy, considerable antibacterial and antibiofilm were demonstrated by propolis aqueous solution on planktonic and mature biofilm of *S. aureus*, with MIC and MBEC values ranging from 2 to 70 μg/mL, while the alcoholic extracts of propolis displayed significantly lower MIC and MBEC values ranged from 2–4 µg/mL on the same strain [72]. The exposure times of biofilm to propolis treatment were different from one study to another. Thus, standard methods for studying biofilm formation and assessing the effectiveness of propolis on biofilm eradication are required. Nonetheless, *S. aureus* biofilms were completely inactivated with 2 μg/mL EEP after 40 h long treatment, indicating that the activity is dependent on treatment times [73].

The biofilm formation of some bacteria is one of the important microbial defence strategies against antibiotics. In this study, double fluorescent staining with resazurin and PI were applied on 24 h-old biofilms to detect the simultaneous effect of propolis on the reductive metabolic activity and the cell viability in the mature biofilm of *S. aureus*. The PI binds specifically to the DNA through the penetration into the cells only with disrupted membranes. This study has clearly shown the concentration-dependent cytotoxic effect of EEP1 on cells within the structure of biofilm. EEP1 significantly decreased the cellular metabolic activity of the four *S. aureus* strains within the biofilm up to 90% at 200 µg/mL (4 MIC value). This result was in good agreement with the 90% elimination of the living *S. epidermidis* cells from the biofilm structure at 4 MIC EEP [74]. Similarly, another group investigated the ethanolic extracts of Brazilian brown propolis (BEEP) on mature biofilms of *S. aureus*, the result showed the reduction of 93% of the viability of the cells present in the biofilms at 125 μg/mL, however, the total biofilm biomass eradication was insignificant [75]. At the concentration of 50 µg/mL (MIC value) of EEP1, MSSA and SA H24 showed higher metabolic activities than that of MRSA and SA H23 (Figure 4), which is in parallel with the resistance presented and the higher thickness of 24 h-old biofilm biomass (Figure 3), indicating a protective effect. The significant decrease in cellular metabolic activity was proportional to the increase in dead cells (Figure 4).

The outcomes of previous research revealed a high efficiency of EEP in the eradication of MSSA biofilms incubated for 24 h at 37 °C, with equal MIC and MBEC values (64–128 µg/mL). It was concluded that the antibiofilm activity of propolis is the most clinically beneficial aspect [57]. The antibiofilm activity of Russian propolis ethanol extracts (RPEE) in the mature biofilm has been reported by Bryan et al. using the MTT assay, it showed a 50% decreased viability of *S. aureus* at a high concentration (5% *w*/*v*) of RPEE. However, at quite high RPEE concentrations (20% *w*/*v*), the confocal and scanning electron microscopy images indicated the complete inactivation of bacterial biofilms after 18 h treatment and demonstrated severe cell wall damage as a possible means of cell lysis [10].

## 3. Materials and Methods

### 3.1. Microorganisms and Culture Conditions

In this study, two Hungarian clinical isolates of MRSA (SA H23 and SA H24) were obtained from the Department of Medical Microbiology, University of Pécs, Hungary. The reference strains methicillin-susceptible *S. aureus* ATCC 29213 (MSSA) and methicillin-resistant *S. aureus* ATCC 700699 (MRSA) were used as negative and positive controls in the susceptibility test, respectively, as well as biofilm positive control strains. All the strains were cultured in tryptic soy broth (TSB) and tryptic soy agar (BDTM, Heidelberg, Germany) with 2% (*w*/*v*) NaCl at 37 °C for 24 h.

### 3.2. Propolis Sources and Extraction

Raw propolis samples were collected during the spring season from six different regions of Hungary including Pécs (P1) [46.0727° N, 18.2323° E], Szombathely (P2) [47.2307° N, 16.6218° E], Szolnok (P3) [47.1621° N, 20.1825° E], Csikóstőttős (P4) [46.3388° N, 18.1591° E], Héhalom (P5) [47.7787° N, 19.5882° E], and Somogybabod (P6) [46.6696° N, 17.7768° E]. The samples were ground, then 100 g of crude propolis samples were mixed with 450 mL of 80% (*v*/*v*) ethanol and incubated in a water bath at 70 °C for 30 min. The ethanolic extracts of propolis (EEP) were filter-sterilized through a 0.22 µm pore size filter (Millipore, Burlington, MA, USA). The final propolis concentration of the stock solutions was 222.2 mg/mL in each sample. The EEP was stored at 4 °C in dark [76].

### 3.3. Determination of the Total Phenolic Content (TPC)

The TPC in the propolis extracts was evaluated indirectly by relating the reducing capacity of propolis and gallic acid standard compound using the Folin-Ciocalteu method [77]. Briefly, 500 µL of 200 µg/mL EEP was mixed with 500 µL of Folin-Ciocalteu reagent (10% *v*/*v*) and 500 µL of Na_2_CO_3_ (2% *w*/*v*). The mixture was incubated in dark at room temperature for 1 h. The absorbance of the reaction mixture was determined at 700 nm against the blank (the reagent mixture without EEP) using a Hitachi U-2910 spectrophotometer (Tokyo, Japan). Gallic acid standard solutions (0.01–0.5 mM) were used for constructing the calibration curve (y = 85.344x − 0.0053; R² = 0.9995). The TPC was expressed as milligram (mg) of gallic acid equivalent (GAE) per gram (g) of propolis dry weight (DW). The chemicals were obtained from Merck Life Science Ltd., Budapest, Hungary. 

### 3.4. Determination of Total Flavonoids Content (TFC)

The flavonoids were determined by the aluminium chloride colorimetric method, as reported by Dias et al. [78]. Briefly, 125 µL of 1 mg/mL EEP was mixed with 625 µL of distilled water and 37 µL of 5% NaNO_2_ solution. After 5 min, 75 µL of 10% AlCl_3_ solution was added, and subsequently, 250 µL of 1 M NaOH and 137 µL of distilled water were added after 6 min to the mixture and well vortexed. The intensity of the pink colour of the reaction mixture was measured at 510 nm against the blank (the same mixture without EEP) using a Hitachi U-2910 spectrophotometer (Tokyo, Japan). Catechin standard solutions (0.022–1.5 mM) were used for constructing the calibration curve (y = 0.6814x + 0.0061, R² = 0.9997). The TFC was expressed as mg of catechin equivalent (CAE) per g of propolis dry weight (DW). The chemicals were obtained from Merck Life Science Ltd., Budapest, Hungary. 

### 3.5. Antibacterial Susceptibility Test

#### 3.5.1. MIC Value Determination

*S. aureus* strains were tested for their susceptibility to EEP1 sample (the sample that was used for all the following experiments, which presented a high level of phenolics and flavonoids contents), oxacillin (Sigma-Aldrich, Darmstadt, Germany), cefoxitin (Sigma-Aldrich, Darmstadt, Germany), and vancomycin (Sigma-Aldrich, Darmstadt, Germany) using broth microdilution method as described by the Clinical and Laboratory Standards Institute [79]. Briefly, the density of the bacterial cells was adjusted to a final concentration of 10^5^ CFU/mL in Muller Hinton broth (MHB) (Biolab, Budapest, Hungary). The cell suspension was mixed in 1:1 ratio with two-fold serial dilutions of EEP1 (12.5–100 µg/mL), oxacillin (0.125–8 µg/mL), cefoxitin, and vancomycin (0.25–16 µg/mL), severally into 96-well cell culture microtiter plates (Costar 3599, Corning, Kennebunk, ME, USA). The concentration of 80% (*v*/*v*) ethanol (solvent of propolis) was kept constant (1%) in each well. The culture was incubated at 35 °C for 20–24 h. The absorbance of the growth was measured at 595 nm using a Thermo Multiskan EX plate reader (Berlin, Germany). The minimum inhibitory concentration (MIC) value was determined as the lowest concentration at which 90% growth inhibition occurred.

#### 3.5.2. Chequerboard Broth Microdilution Method

The broth microdilution chequerboard method was used to study the possible synergistic effect between EEP1 with selected antibiotics (oxacillin, cefoxitin, and vancomycin). Briefly, 100 μL of bacterial suspension adjusted to 10^8^ CFU/mL was distributed into a 96-well microtiter plate (Costar 3599, Corning, Kennebunk, ME, USA) containing 50 µL of two-fold serial dilutions of EEP1 (3.13–400 µg/mL) and 50 µL of selected antibiotics (0.03–16 µg/mL). The plate was incubated at 35 °C for 20–24 h. The absorbance of the growth was measured at 595 nm using a Thermo Multiskan EX microtiter plate reader (Berlin, Germany). A calculation matrix was created to convert the absorbance to percentages of the growth. The type of interaction between the EEP1 and the selected antibiotics was defined by the calculation of the fractional inhibitory concentration index (FICI). The FICI was computed according to the following equations:

FICI = (MIC value of the selected antibiotic in combination/MIC value of the selected antibiotic alone) + (MIC value of EEP1 in combination/MIC value of EEP1 alone).

The combination effect of antibiotics with EEP1 was considered, as synergistic when FICI ≤ 0.5, an additive when 0.5 < FICI < 1, indifferent when 1 ≤ FICI < 4, and antagonistic when FICI > 4 [19].

### 3.6. Biofilm Formation and Quantification Assay

For testing the effect of EEP1 on the biofilm of *S. aureus*, the cell number of an exponential-phase culture was adjusted to 10^3^ cells/mL into TSB supplemented with 0.25% (*w*/*v*) glucose. The cell suspension was treated with two-fold serial dilutions of EEP1 (12.5–200 µg/mL) in the final volume of 200 µL (1:1, *v*/*v*) into 96-well microtiter plates (Sarstedt, REF 833934500, Numbrecht, Germany), and incubated at 37 °C for 24 h. The cells were washed three times with 200 µL sterile PBS (pH 7.2), Then it was left to dry at room temperature. The biofilm was fixed with 100 µL of 99% (*v*/*v*) methanol for a 15 min incubation. For the quantification of biofilm biomass, the dried biofilm was stained with 200 µL of 0.13% (*w*/*v*) crystal violet for 15 min. The unbound dye was removed by washing three times with 200 µL of PBS. The crystal violet dye was eluted with 200 µL of 33% acetic acid glacial to solubilize the biofilm-bound dye by incubating for 15 min. The absorbance of biofilm biomass was measured at 595 nm using a Thermo Multiskan EX plate reader (Berlin, Germany). The absorbance of an inoculated well without propolis treatment served as a positive control and the absorbance of an uninoculated well served as a negative control. The minimum biofilm inhibitory concentration (MBIC) was defined as the lowest concentration that inhibited at least 90% biofilm formation. Then the cut-off value (ODc) was established; ODc = average OD of negative control + (3 × SD of negative control); OD = average OD of a strain subtracted from ODc. For the interpretation of the results, strains were divided into the following categories: OD ≤ ODc = not biofilm-former, ODc < OD ≤ 2 × ODc = weak biofilm-former, 2 × ODc < OD ≤ 4 × ODc = moderate biofilm-former, 4 × ODc < OD = strong biofilm-former [80].

### 3.7. Biofilm Eradication Assay

The ability of EEP1 to eradicate the 24 h-old biofilm of *S. aureus* was determined as previously mentioned in the biofilm formation and quantification assay (See Section 3.6). The cell suspension was incubated for 24 h at 37 °C without EEP. Then, the supernatants were removed and the wells were treated with EEP1 (12.5–200 µg/mL) for 16 h at 37 °C. The planktonic cells were discarded and only the tightly attached biofilm was stained with crystal violet, resazurin, and propidium iodide (PI) to quantify the biofilm biomass, metabolic activity, and cellular death, respectively. The crystal violet assay for the quantification of biofilm biomass was mentioned in Section 3.6. To quantify the metabolic activity of the cells present within the mature biofilm, the wells were labelled with 1 µM resazurin solution (200 μL) in dark for 30 min. The metabolic activity is proportional to the rate of resazurin reduction that was determined by measuring the fluorescence at (λ_Ex/Em_ = 560/590 nm). The dead cells were determined by treating the mature biofilm with 200 μL of 20 μM PI in the dark for 15 min. The PI is an intercalating fluorescent agent, binding of PI to DNA causes a redshift of the excitation maximum to 540 nm and the emission maximum to 640 nm. The fluorescence measurements were determined using a PerkinElmer EnSpire multimode plate reader (Auro-Science Consulting Ltd., Budapest, Hungary). The fluorescence values for resazurin and PI were converted to percentages. Then, the percentage was calculated by supposing the positive control as 100% metabolically active cells (fluorescence of the cells in the presence of EEP1/fluorescence of the cells in the absence of EEP1 × 100) and 0% dead cells ([fluorescence of the cells in the presence of EEP1/fluorescence of the cells in the absence of EEP1 × 100] − 100). Furthermore, the minimum biofilm eradication concentration (MBEC_50_) was computed as the lowest concentration that eradicate at least 50% of biofilm.

### 3.8. Statistical Analyses

All experiments were carried out in triplicates and repeated three times, the data were presented as mean ± standard deviation (SD). Multiple comparisons between the groups were analysed with one-way ANOVA followed by Tukey’s test. Differences between the means of samples were considered significant when *p* < 0.05 (* *p* < 0.05, ** *p* < 0.01, and *** *p* < 0.001). Statistical analysis and graphing were conducted by OriginPro software (version 2016, OriginLab Corporation, Northampton, MA, USA).

## 4. Conclusions

The emergence of resistant strains has stimulated the discovery of new therapeutic agents, such as propolis, which is considered as an effective natural product showing a wide array of biological properties including antimicrobial activities, and other pharmaceutical applications. This study illustrated the variation of Hungarian propolis samples in terms of total phenolic and flavonoid contents. The EEP1 sample revealed the highest concentration of phenols and flavonoids, which resulted in antibacterial sensitivity to all the studied strains. The Hungarian propolis extract showed promising efficacy in declining the planktonic cell growth, and degrading the biofilm of clinical MRSA isolates. Thus, suggesting that EEP1 has the ability to eradicate the *S. aureus* mature biofilm. Further, our findings indicated that propolis extract is able to inactivate the metabolic activity of *S. aureus* strains within the biofilm and causes the cell death. However, the interactions of EEP with antibiotics need further investigation for the in vivo application. The biological potentials and mechanisms of action of propolis could be an ideal candidate for the development of new therapy, cost-effective antimicrobial, and antibiofilm agents.

## Figures and Tables

**Figure 1 molecules-27-00574-f001:**
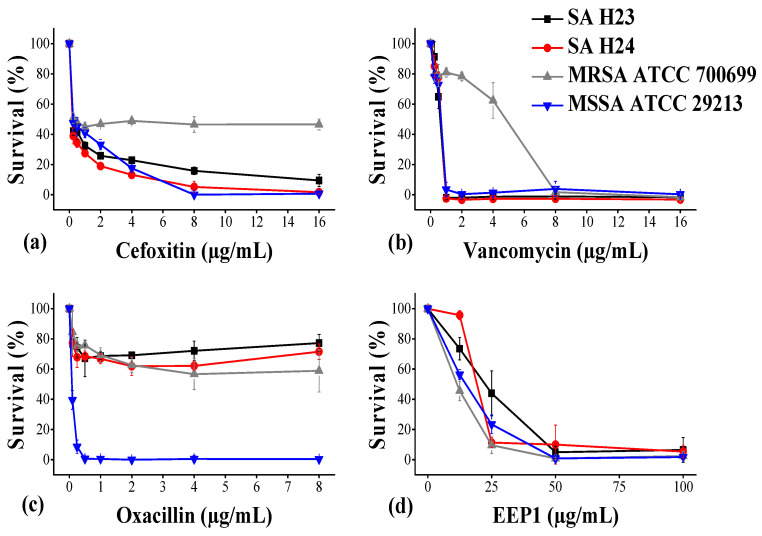
Inhibitory effect of antibiotic drugs: (**a**) Cefoxitin (0.25–16 µg/mL); (**b**) Vancomycin (0.25–16 µg/mL); (**c**) Oxacillin (0.125–8 µg/mL); and (**d**) EEP1 (12.5–100 µg/mL) on *S. aureus*; clinical isolates (SA H23 and SA H24), and reference strains methicillin-susceptible *S. aureus* ATCC 29213 (MSSA) and methicillin-resistant *S. aureus* ATCC 700699 (MRSA).

**Figure 2 molecules-27-00574-f002:**
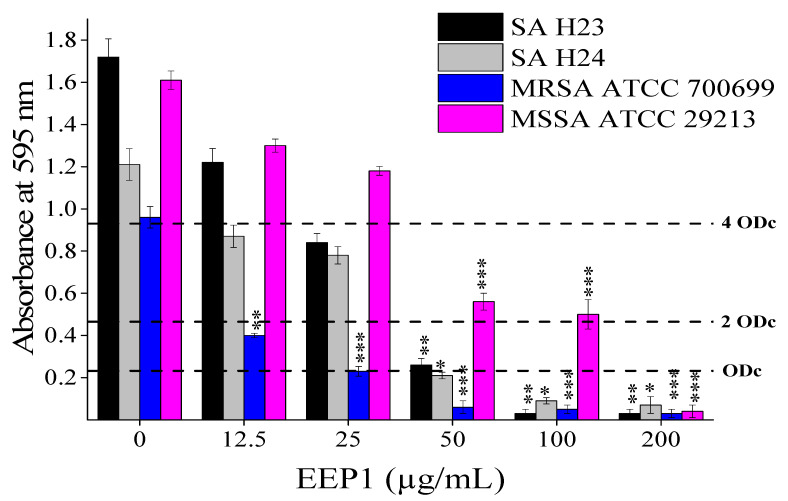
The effect of EEP1 (12.5–200 µg/mL) on the prevention of biofilm formation of *S. aureus* SA H23 and SA H24 clinical isolates, and *S. aureus* ATCC 29213 (MSSA) and *S.aureus* ATCC 700699 (MRSA) strains after 24 h incubation at 37 °C. (ODc) the cut-off value of the optical density. (OD ≤ ODc) means no biofilm, (ODc < OD ≤ 2 ODc) means weak biofilm, (2 ODc < OD ≤ 4 ODc) means moderate biofilm, (4 ODc < OD) means strong biofilm. Asterisks indicate statistically significant differences between each treatment of EEP1 and in absence of EEP1 (* *p* <0.05, * *p* < 0.01, *** *p* < 0.001).

**Figure 3 molecules-27-00574-f003:**
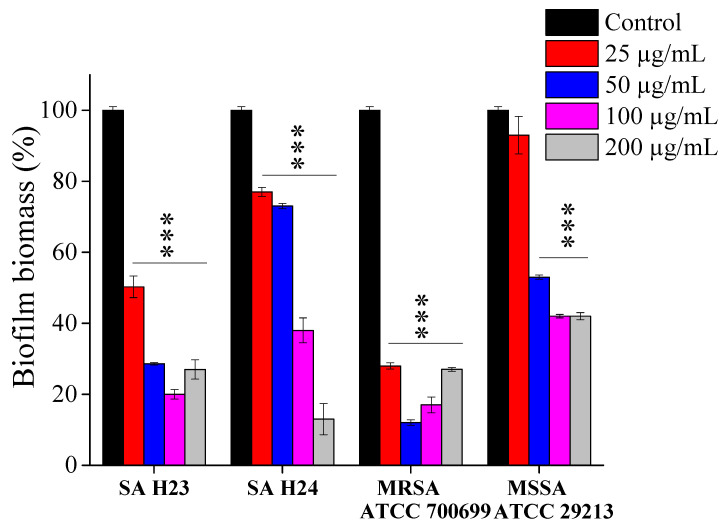
The eradication effect of EEP1 on 24 h-old biofilm biomass of *S. aureus* SA H23, SA H24 clinical isolates, *S. aureus* ATCC 29213 (MSSA), and *S. aureus* ATCC 700699 (MRSA) by crystal violet staining. Asterisks indicate statistically significant differences (one-way ANOVA) between each concentration treatment of EEP1 and untreated biofilms (*** *p* < 0.001).

**Figure 4 molecules-27-00574-f004:**
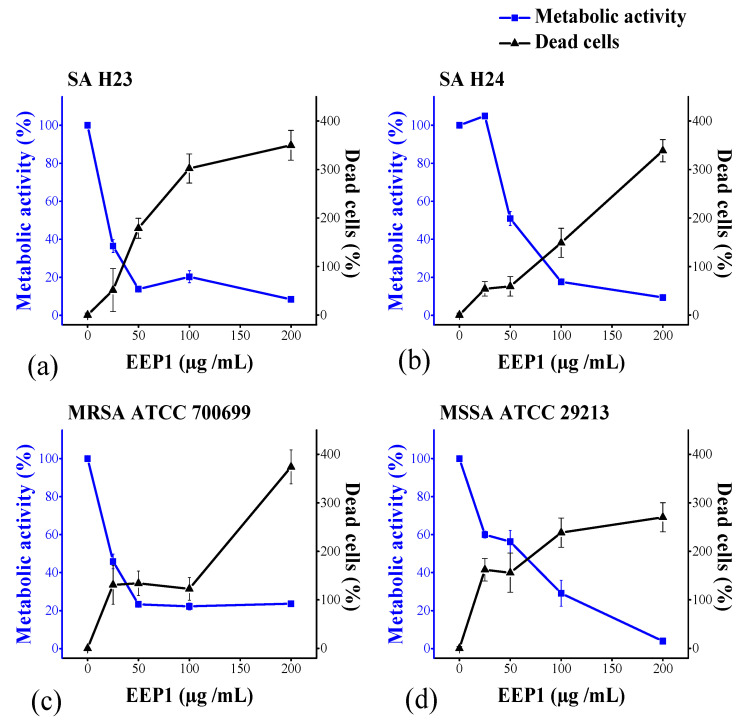
The effect of EEP1 (25–200 µg/mL) on 24 h-old biofilms formed by: (**a**) Clinical isolates of *S. aureus* SA H23, (**b**) SA H24, (**c**) *S. aureus* ATCC 700699 (MRSA), and (**d**) *S. aureus* ATCC 29213 (MSSA), after 16 h incubation via the determination of metabolic activity of the cells that is in proportion to the resazurin fluorescent reaction (blue lines) and dead cells that is proportional to the fluorescence of propidium iodide (black lines) within the mature biofilms.

**Table 1 molecules-27-00574-t001:** Concentrations of phenols and flavonoids in dry weight (DW) of six propolis samples.

Propolis Samples	TPC (mg GAE/g DW)	TFC (mg CAE/g DW)
EEP1	71.1 ± 4.3 ^a^	273.2 ± 10.2 ^a^
EEP2	55.8 ± 2.0 ^b^	172.8 ± 11.5 ^b^
EEP3	47.9 ± 0.2 ^c^	164.1 ± 2.7 ^b,c^
EEP4	44.0 ± 0.5 ^c^	142.5 ± 4.2 ^c,d^
EEP5	34.6 ± 1.0 ^d^	147.3 ± 12.9 ^c,d^
EEP6	10.4 ± 1.4 ^e^	33.8 ± 3.0 ^e^

^a–e^ Different letters indicate significant differences between the regions within the same column (*p* < 0.05). Values represent mean ± standard deviations (*n* = 3).

**Table 2 molecules-27-00574-t002:** The MIC value of EEP1 alone and in combination with antibiotics on *S. aureus* (inoculum of 10^8^ CFU/mL); clinical isolates (SA H23 and SA H24), reference strains methicillin-susceptible *S. aureus* ATCC 29213 (MSSA), and methicillin-resistant *S. aureus* ATCC 700699 (MRSA), and the type of interaction according to fractional inhibitory concentration index (FICI), (N) not calculated.

*S. aureus* Strains	Drugs	MIC Value (µg/mL)		FICI (Type of Interaction)
		Alone	In Combination	
SA H23	EEP1	200	<200	N
	Oxacillin	<4	<4	
	EEP1	200	200	N
	Cefoxitin	<8	0.13	
	EEP1	200	3.13	0.03 (synergistic)
	Vancomycin	2	<200	
SA H24	EEP1	200	200	N
	Oxacillin	<4	0.06	
	EEP1	200	200	N
	Cefoxitin	<8	0.13	
	EEP1	200	3.13	0.03 (synergistic)
	Vancomycin	2	0.03	
MSSA	EEP1	100	25	0.27 (synergistic)
	Oxacillin	4	0.06	
	EEP1	100	3.13	0.05 (synergistic)
	Cefoxitin	2	0.03	
	EPP	100	3.13	0.05 (synergistic)
	Vancomycin	2	0.03	
MRSA	EEP1	400	400	N
	Oxacillin	<4	0.06	
	EEP1	400	200	N
	Cefoxitin	<8	0.13	
	EPP1	400	6.25	0.05 (synergistic)
	Vancomycin	4	0.13	

## Data Availability

The data presented in this study are available on request from the Department of General and Environmental Microbiology, Institute of Biology, University of Pécs.

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
