# Peer review of "Evaluation of Total Phenolic and Flavonoid Contents, Antibacterial and Antibiofilm Activities of Hungarian Propolis Ethanolic Extract against Staphylococcus aureus"

_molecules, 2022, doi:10.3390/molecules27020574_

Round 1
Reviewer 1 Report
Manuscript "Antimicrobial and antibiofilm activities of Hungarian propolis ethanolic extract against methicillin-resistant and methicillin-sensitive Staphylococcus aureus" presents interesting research results and an important topic, which is the use of natural substances, e.g. propolis, as an anti-bacterial substance.
Detailed comments:
Line 63 - supplement the review with the possibility of using propolis in various industries, e.g. in the food industry doi: 10.1111 / ijfs.14753
Table 1 - the statistical analysis of the results is missing. Standard deviations are very large, the authors should re-analyze the obtained results, because standard deviations should not exceed 10% of the average result.
The results are correctly described, while the discussion of the divisions is very small, and it is good for the authors to expand it.
The methodology is well written and replicable.
In Conclusion, it would be worth adding more detailed research results to summarize the results obtained.
The References chapter requires adjustment to the journal's requirements.
Reviewer 2 Report
This work is useful for reader in the field. Previous studies with propilis reported synergistic of propilis with honey and other extracts. They reported with their chemical compounds. This work is similar to previous reports. Therefore, mode of action using some techniques such as flow cytometry must be performed in order to understand their mechanism which makes this work new and more useful for readers. If, not it is lack of novelty.
Reviewer 3 Report
In this manuscript, Kocsis et al reported the detail investigation of the biological properties of the ethanolic extract of Hungarian propolis. They have successfully tested the antimicrobial and antibiofilm properties of the propolis against methicillin-resistant and methicillin-sensitive Staphylococcus aureus. Currently, antimicrobial resistance in bacteria has become a grave challenge which has significantly increased morbidity and mortality rates in antibacterial therapy. Particularly, multidrug resistance has been observed both in Gram-positive and -negative bacteria which is difficult to treat with conventional antibiotics. Therefore, finding strategies against the development of antibiotic resistance is a priority for the life sciences community and for public health. Herein, the authors have successfully evaluated the antimicrobial and antibiofilm activities of ethanolic extract of propolis (EEP) on methicillin-resistant and sensitive Staphylococcus aureus. All the experiments are nicely performed and the results have also been explained in detail, therefore this manuscript is acceptable for publication after some minor revisions.
Few languages changes are required, as some mistakes were found in sentence construction, use of prepositions and tense and some spelling mistakes has been detected at some places.
The authors claim that the antibacterial properties of propolis is related to the total phenolic and flavonoid contents which is indirectly dependent on various factors including region, time and other climatic conditions. Since the authors have collected several propolis samples from different regions of Hungary, did they find any correlation between the antibacterial properties of all the collected samples and their polyphenolic contents against Staphylococcus aureus.
Secondly, the authors should also highlight the reasons behind the huge difference in the polyphenolic contents of various propolis samples
Conclusion is week, more results should be summarized in this section.
Round 2
Reviewer 1 Report
The corrections have been made in the text, I have no more comments.
Author Response
Thank you for your positive response.
Reviewer 2 Report
The authors can not add more experiment such as flow cytometry at this time, hopw you cancomplete the spectrum of the measured parameters with further analyses in the future.
Author Response
Thank you for your positive response.